# Individual, community, and societal drivers of adolescent peer violence in Asella, Ethiopia: A school-based cross-sectional study utilizing the socio-ecological model

Ayalneh Demissie[1]*, Tesfa G/Meskel[2], Kasim Kimo[3], Muhammedawel Kaso[1], Yehwalishet Demeke[4], Tahir Aman[5], Yonas Mulugeta[6], Addis Wordofa[1]

1 Department of Public Health, College of Health Sciences, Arsi University, Asella, Ethiopia, 2 Department of Pediatrics, College of Health Sciences, Arsi University, Asella, Ethiopia, 3 Department of Psychology, College of Social Sciences, Arsi University, Asella, Ethiopia, 4 College of Business and Economics, Arsi University, Asella, Ethiopia, 5 Department of Anaesthetics, College of Health Sciences, Arsi University, Asella, Ethiopia, 6 Department of Biomedical Sciences, College of Health Sciences, Arsi University, Asella, Ethiopia

* ayalnehdemissis@gmail.com

## Abstract

### Background

Adolescent peer violence is a critical public health challenge in low- and middle-income countries. However, existing research in sub-Saharan Africa often focuses disproportionately on intimate partner violence or male-centered aggression, leaving a significant data gap regarding broader peer-to-peer dynamics in rapidly urbanizing Ethiopian settings. Furthermore, there is an urgent need for evidence that integrates psychosocial traits with environmental influences. This study addresses these gaps by utilizing the **Socio-Ecological Model (SEM)**—a framework that examines the interplay between individual, relationship, community, and societal factors—to investigate peer violence in Asella, Ethiopia. We specifically explore the role of **General Self-Efficacy (GSE)**, defined as an individual's belief in their ability to perform across a variety of situations, as a potential driver of behavior. Our objective was to determine the magnitude of violence victimization and perpetration and to identify multi-level risk factors among school-going adolescents.

### Methods

A school-based cross-sectional study was conducted among **542 adolescents** (response rate = 99.5%) using a structured, self-administered questionnaire. Magnitude was assessed through standardized behavioral scales, and psychological resilience was measured using the General Self-Efficacy (GSE) scale. Multivariable binary logistic regression was utilized to identify independent predictors across the Socio-Ecological Model (SEM) levels.

**Data availability statement:** All relevant data are within the paper and its Supporting Information file.

**Funding:** This research was supported by Arsi University (Award Number: CoHS/R/10/2024/2025, received by AD), which provided financial support for data collection and field per diems. The funders had no role in study design, data collection and analysis, decision to publish, or preparation of the manuscript. No authors received a salary from any funders for this specific work.

**Competing interests:** The authors declare no conflicts of interest.

**Abbreviations:** AOR: Adjusted Odds Ratio; CI: Confidence Interval; GSHS: Global School-based Health Survey; HHCs: Household Challenges; H-L test: Hosmer–Lemeshow test; P-value: Probability Value; SD: Standard Deviation; SEM: Socio-Ecological Model; SPSS: Statistical Package for the Social Sciences; SRS: Simple Random Sampling; YRBS: Youth Risk Behavior Surveillance System.

## Results

The prevalence of violence **victimization was 41.6%** (95% CI: 37.3–45.9%) and **perpetration was 33.8%** (95% CI: 29.7–38.1%). Based on bullying roles, 15.0% were "Bully-Victims," who reported the lowest mean self-efficacy score (25.5 ± 3.6). Multivariable analysis revealed that **low self-efficacy (≤28)** was the strongest predictor for victimization (**AOR = 2.88**; 95% CI: 1.95–4.26), while **witnessing community violence** was the primary driver for perpetration (**AOR = 2.52**; 95% CI: 1.77–3.59). Male gender and belonging to the lowest wealth quartile significantly increased the odds for both outcomes.

## Conclusions

Adolescent violence in Asella is a multi-dimensional public health issue driven by a cycle of low individual agency and environmental normalization of aggression. Strategies to mitigate this burden must move beyond the individual level. We recommend a comprehensive response that integrates school-based self-efficacy enhancement, community-level peace-building initiatives to reduce the impact of witnessed violence, and societal-level poverty alleviation. Such a multi-level approach is essential to disrupt the interplay between psychological vulnerability and environmental risk.

## 1. Introduction

**Background/Rationale** Violence involvement, encompassing both perpetration and victimization, represents a critical global public health concern with profound effects on adolescent developmental trajectories, mental health, and academic outcomes [1–4]. In Sub-Saharan Africa (SSA), the magnitude of violence exposure remains alarmingly high [5–7], yet regional research often neglects the experiences of middle adolescents (ages 14–19) and frequently fails to adequately assess the interplay of risk and protective factors within a comprehensive, multi-level framework [8–10].

The complex etiology of adolescent violence is best understood through the lens of the **Socio-Ecological Model (SEM)** [11]. The SEM posits that violence results from the interaction of factors at various levels: the **Individual** (e.g., self-efficacy, gender) [12,13], the **Relationship** (e.g., family structure, Adverse Childhood Experiences or HHCs) [14–16], the **Community** (e.g., neighbourhood violence exposure) [9], and the **Societal** (e.g., economic inequality, rigid gender norms) [11,17]. This model allows for the identification of both psychological vulnerabilities and environmental drivers of violence. Violence exposure in adolescents, including victimization and witnessing, is associated with elevated risk of anxiety, depression, scholastic difficulties, and conduct problems [2,7,18].

In applying this framework, it is critical to distinguish between peer violence and Gender-Based Violence (GBV), as they often overlap but stem from different structural drivers. In this study, **peer violence** is defined as physical, psychological, or relational aggression occurring between students of similar age and social status

within the school environment. While peer violence can include gendered elements, **GBV** specifically refers to harmful acts directed at an individual based on their gender, rooted in structural inequalities and power imbalances. While much of the existing literature in Ethiopia has focused on GBV—particularly female-targeted domestic abuse—there is a significant lack of real-time, school-based data addressing the broader dynamics of peer-to-peer aggression in rapidly urbanizing settings like Asella.

This study addresses a critical gap noted by Pells and Morrow (2018) [19], who emphasize the need for more understanding of the nature and dynamics of violence among children in Ethiopia. By investigating multi-level predictors of violence involvement, this research provides vital, context-specific data on a problem that carries enormous social and economic costs [20–22]. Specifically, this study aims: 1) to determine the magnitude of violence victimization and perpetration among school adolescents in Asella, and 2) to identify the risk and protective factors associated with violence victimization and perpetration, explicitly applying the Socio-Ecological Model to isolate individual, relationship, community, and societal influences.

## 2. Methods

### Study design

The study employed a school-based, quantitative cross-sectional design.

### Setting

The study was conducted in Asella city, Oromia Regional State, Ethiopia. Asella is the administrative center of the Arsi Zone and is located approximately 175 kilometers southeast of Addis Ababa. The study population included adolescents enrolled in both public and private secondary schools (Grades 9–12) within the town. During the 2023 academic year, the total student enrolment for these grades was approximately 13,411. Data collection was carried out from April 15/ 2025 to May/15/ 2025.

### Participants

**Source and study population.** The source population consisted of all adolescents enrolled in public and private secondary schools (**Grades 9–12**) in Asella, Ethiopia. To ensure a developmentally consistent sample of middle adolescents, the study population was restricted to those aged **15–19 years** within the randomly selected school sections who were available during the data collection period.

**Eligibility criteria:**

• **Inclusion Criteria**\* Adolescents aged **15–19 years** enrolled in Grades 9–12.

• All students present in the selected schools during the data collection period.

• Students who provided informed assent (accompanied by parental consent or opt-out notification as per the Institutional ERC protocol).

**Exclusion criteria:** \* Students who were acutely ill at the time of data collection.

• Students who had severe hearing or speech impairments, or known severe cognitive impairments that prevented them from independently participating in the survey.

### Sampling procedure

A multi-stage proportionate stratified random sampling procedure was employed. Four secondary schools were selected at random. The required sample size (N = 545) was then allocated to each school and grade level proportional to the

student population size. The required number of adolescents was then selected using Simple Random Sampling (SRS) from the sampling frame (class rosters) of eligible students.

## Variables

Variables were structured according to the multi-level framework of the Socio-Ecological Model (SEM) [23–25] to identify factors influencing adolescent behavior across different domains:

**Dependent variables:**

- **Violence Victimization:** (Binary: Yes/No)

- **Violence Perpetration:** (Binary: Yes/No)

**Independent variables (Socio-Ecological Levels):**

- **Individual Level:** Gender (Male/Female), Age (Continuous/Categorical), and General Self-Efficacy (GSE).

- **Relationship Level:** Parental Status (Intact/Non-intact household) and Adverse Childhood Experiences (HHCs).

- **Community Level:** Witnessing Community Violence (Exposure to fights or weapon use in the neighbourhood).

- **Societal Level:** Household Wealth (Categorized into quartiles).

- **Additional Socio-Demographics:** Ethnicity, Educational Level (Grade), and Family Size.

## Data sources/Measurement

Data were collected using a **structured, pre-tested, self-administered questionnaire**. This format was chosen specifically to ensure participant privacy and to facilitate the disclosure of sensitive behaviors related to violence and victimization. The tool was adapted from the **Global School-Based Student Health Survey (GSHS)** [26] and other validated instruments [24,25].

## Instrument validity and reliability

We utilized standardized instruments adapted to the local context. Table 1 summarizes the conceptual framework and psychometric properties of these measures. **Violence victimization** (15 items) and **perpetration** (9 items) were assessed

Table 1. Conceptual framework: Hypothesized pathways to peer violence outcomes, Asella-2025.

| SEM level | Variable | Measurement / Instrument | Response options / Coding | Notes / Psychometrics |
|---|---|---|---|---|
| **Dependent** | **Violence Victimization** | School Life Survey (15 items) [27] | Yes / No; Defined as ≥1 act in past six months | Cronbach's α = 0.83; 1-week test-retest = 0.94 |
| | **Violence Perpetration** | School Life Survey (9 items) [27] | Yes / No; Defined as ≥1 act in past six months | Cronbach's α = 0.83; 1-week test-retest = 0.84 |
| | **Bullying Roles** | Derived from victimization & perpetration [27] | Victim Only, Perpetrator Only, Bully-Victim, Non-Involved | — |
| **Individual** | **Self-Efficacy** | General Self-Efficacy Scale (10 items, 4-point Likert) [28] | Total Score 10–40. Categorized as Low (≤28) vs. High (>28) | Cronbach's α > 0.70 |
| **Relationship** | **Parental Status** | Self-report [27] | Categorized as Intact (Married/Living together) or Non-Intact | — |
| **Community** | **Witnessing Community Violence** | Self-report (Dichotomous) [29] | Yes / No | — |
| **Societal** | **Household Wealth** | Composite index via Principal Component Analysis (PCA) [27] | Quartiles: Lowest, Second, Third, Highest | — |

using the School Life Survey [27], which demonstrated high internal consistency (Cronbach's α = 0.83 for both) and strong 1-week test-retest reliability (r = 0.94 and 0.84, respectively).

Individual psychological agency was measured using the **General Self-Efficacy (GSE) scale** [28]. The scale showed acceptable reliability (α > 0.70) in this population. To ensure cultural and linguistic validity, all instruments underwent a rigorous translation and back-translation process into Afan Oromo and Amharic, followed by a pre-test on 5% of the sample to ensure clarity and conceptual equivalence.

- **Violence Assessment:** Participants reported incidents of physical and psychological aggression (victimization and perpetration) occurring within the **last 12 months**.

- **General Self-Efficacy (GSE):** Assessed using the 10-item GSE scale, which measures an adolescent's perceived agency and ability to cope with stressors.

- **Socio-Economic Status:** Household wealth was measured through a checklist of assets and analyzed using **Principal Component Analysis (PCA)**.

**Conceptual framework**

The conceptual framework for this study is derived from the **Socio-Ecological Model (SEM)** [11] of violence prevention. As shown in Fig 1 this framework structures the analysis of independent variables (risk and protective factors) and their influence on the dependent outcomes (Victimization and Perpetration).

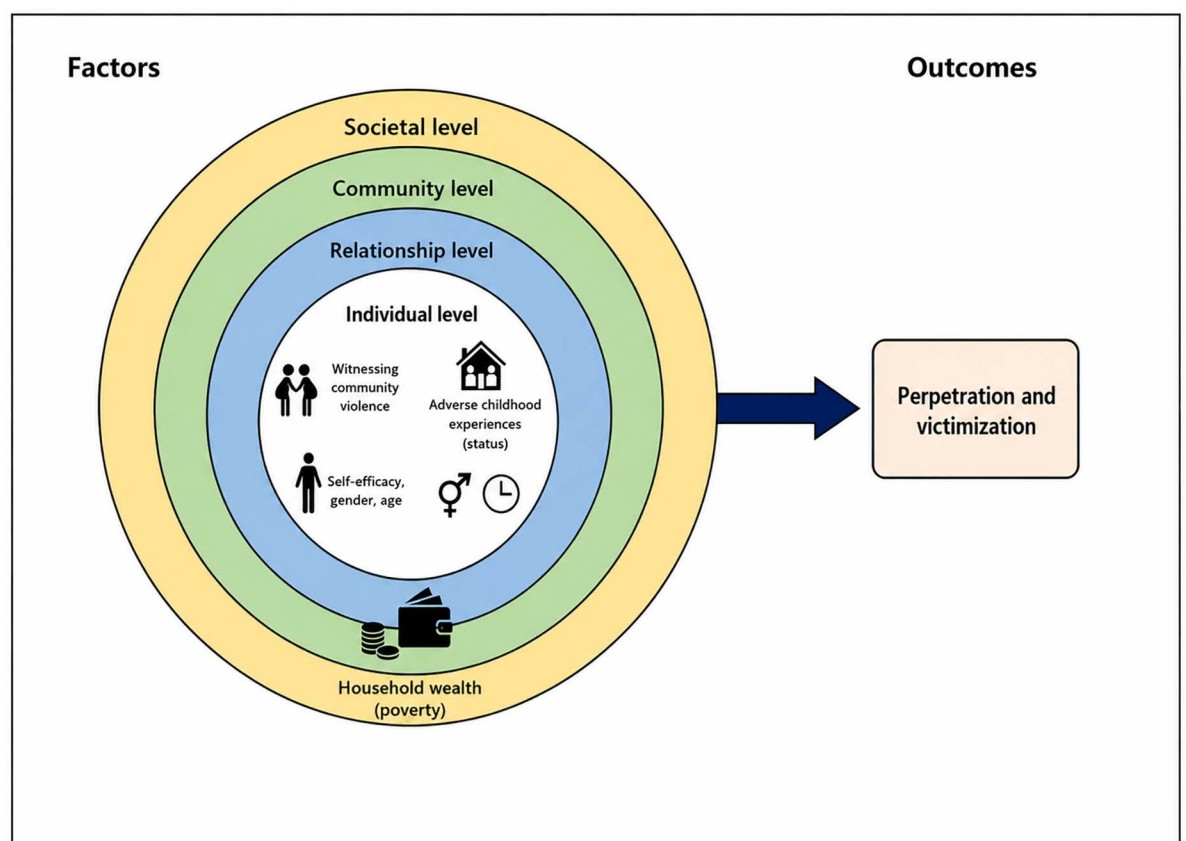

**Fig 1. Ecological factors and their influence on violence outcomes, Asella, Ethiopia, 2025.**

The model emphasizes that behavior is influenced by multiple, nested levels of environmental and individual factors. The central pathway follows the trajectory:

**Ecological factors → Violence outcomes (Perpetration & Victimization)**

The concentric circles represent the four levels of the SEM, starting with the individual and moving outward. Factors at the inner levels are influenced by factors at the outer levels. Table 1 details the specific variables and their psychometric properties.

### Bias

In accordance with **STROBE Item 9**, the following measures were implemented to control for bias:

- **Social Desirability and Self-Reporting Bias:** To minimize the risk of adolescents' under-reporting violent behaviors or over-reporting "positive" traits, the survey was **self-administered**. Participants completed the questionnaires in a private setting where their responses could not be seen by teachers, peers, or researchers.

- **Selection Bias:** Minimized through **Simple Random Sampling (SRS)** of classroom sections from the total population of 13,411 students, ensuring every student had an equal chance of being included in the study.

- **Recall Bias:** The study focused on events within a **12-month recall period**, providing a balance between capturing meaningful data and the limitations of adolescent memory.

- **Instrument Bias:** A **pre-test** was conducted on 5% of the sample in a similar setting to ensure that the translated questions were clear and culturally appropriate before the main data collection phase.

### Sample size

The sample size was calculated using **Epi Info™ version 7.1.5**. The calculation was based on an assumed prevalence of violence among school-attending adolescents of **75%** [30], a **95% Confidence Interval**, and a **5% margin of error**. To account for the multi-stage sampling approach, a **Design Effect (DE) of 1.5** was applied. This resulted in a total required sample size of **545**. Following data collection and cleaning, the final analysis included **542** complete responses (99.5% response rate).

### Quantitative variables

To facilitate interpretation relative to reference groups in the logistic regression analysis, all quantitative independent variables were categorized:

- **Age:** Dichotomized into younger adolescents (**14–15 years**) and older adolescents (**≥16 years**) [26].

- **Self-efficacy:** Categorized as **Low (≤28)** versus **High (>28)**, based on the sample's calculated mean score [28].

- **Household Wealth:** Analyzed as a categorical variable using **Quartiles** (Lowest to Highest) derived from the Principal Component Analysis (PCA) of household assets [27].

### Statistical methods

Data were checked for completeness and consistency, then entered, cleaned, standardized, and managed using Epi-Data Manager v4.6 [25], and exported to SPSS v27.0 for analysis. Descriptive statistics (frequencies, percentages, means ± SD) characterized the sample [27,29]. Initial associations were screened using the Chi-Squared ($\chi2$) Test. Differences in mean General Self-Efficacy scores across the four bullying roles were tested using a **One-Way Analysis of Variance (ANOVA)**, followed by **Tukey's HSD post-hoc comparisons** when significant differences were found [27].

For the primary objective, two separate multivariable binary logistic regression models identified independent predictors for victimization and perpetration. Variables showing an association at P < 0.25 in bivariable analysis were included. Model diagnostics assessed multicollinearity (VIF < 5) and overall fit (Hosmer-Lemeshow Test). Results are reported as Adjusted Odds Ratios (AOR) with 95% Confidence Intervals (CIs). Statistical significance was defined as P < 0.05.

### Ethical considerations and consent

**Ethical approval**: Ethical clearance was obtained from the Ethical Review Committee (ERC) of Arsi University, College of Health Sciences (**Protocol Number**: **CoHS/R/10/171/2024/2025**). The study was conducted in accordance with the Declaration of Helsinki. Permission to conduct the study was further secured from the Asella City Administration Education Bureau and the respective school administrations.

**Informed Consent** Participants were provided with a clear explanation of the study's purpose, the voluntary nature of participation, and their right to withdraw at any time without penalty.

- **For participant's ≥18 years:** Formal written informed consent was obtained directly from the adolescents.

- **For participants <18 years:** Written informed assent was obtained from the adolescent, along with written informed consent from their parents or legal guardians.

**Privacy and Data Protection** to ensure confidentiality, all data collection was anonymous. No personal identifiers (such as names or specific addresses) were collected. Participants completed the self-administered questionnaires in a private classroom setting to minimize peer influence and ensure privacy. All physical data were kept in a locked cabinet, and digital data were stored on a password-protected server accessible only to the research team.

## 3. Results

### Demographic and socio-economic characteristics

The study included a total of **542** adolescents, achieving a response rate of **99.5%**. The mean age of the participants was **17.4 years (SD = 1.2)**, with ages ranging from 14 to 19 years. Gender was nearly equally distributed, with **279 (51.5%)** participants being female and **263 (48.5%)** being male.

The majority of students **(n = 421, 77.7%)** lived in intact households (parents married or living together). Regarding economic status, the sample was distributed across wealth quartiles, with **119 (22.0%)** belonging to the lowest wealth group for the detail see Table 2.

### Violence and self-efficacy summary by bullying role

#### Outcome data

**The "Bully-Victim" Paradox.** While 41.6% were victims and 33.8% were perpetrators, the most important finding is the **15.0% "Bully-Victim" group**. This group is the most vulnerable because they are trapped in a cycle of both receiving and giving aggression. By relating this to their **lowest mean self-efficacy (25.5)**, the study suggests that their violence is **reactive**.

Unlike "Pure Bullies" (who have higher self-efficacy, 30.5), Bully-Victims lack the emotional regulation and confidence required to resolve conflicts peacefully, so they lash out.

### Self-efficacy as a protective factor

By looking at the **Non-Involved group (67.0%)**, who have the **highest self-efficacy (31.1)**, we establish a "baseline for resilience." This relates the psychological data to the outcome data by showing that high self-efficacy likely acts as a "buffer" or protective factor that keeps the majority of students from getting involved in violence at all as shown in Table 3.

**Table 2. Sociodemographic characteristics of study populations in Asella, 2025 (N = 542).**

| Characteristic | Category | N | % |
|---|---|---|---|
| Gender | Male | 263 | 48.5 |
| | Female | 279 | 51.5 |
| Age (years) | Mean (SD) | 17.4 (1.2) | — |
| | Min–Max | 14–19 | — |
| Religion | Orthodox Christian | 324 | 59.8 |
| | Muslim | 138 | 25.5 |
| | Protestant | 75 | 13.8 |
| | Other (Catholic/Traditional) | 5 | 0.9 |
| Parental Status | Married / Living together | 421 | 77.7 |
| | Separated / Divorced / Widowed | 118 | 21.8 |
| | Never married | 3 | 0.6 |
| Wealth Quartile | Lowest (Poorest) | 119 | 22.0 |
| | Second | 141 | 26.0 |
| | Third | 147 | 27.0 |
| | Highest (Richest) | 136 | 25.0 |

**Table 3. Violence magnitude and mean GSE scores by bullying role (N = 542).**

| Outcome / Bullying Role | % | 95% CI | Mean GSE Score (SD) |
|---|---|---|---|
| **Overall Victimization** | 41.6 | 37.3–45.9% | — |
| **Overall Perpetration** | 33.8 | 29.7–38.1% | — |
| **Non-Involved** | 67.0 | 63.0–71.0% | 31.1 (±4.2) |
| **Pure Victim** | 10.0 | 7.6–12.8% | 26.2 (±3.8) |
| **Pure Bully** | 8.0 | 5.9–10.6% | 30.5 (±4.5) |
| **Bully-Victim (Both)** | 15.0 | 12.1–18.3% | 25.5 (±3.6) |

## Statistical validation via Tukey's HSD

The post-hoc tests are the "bridge" between these findings. Because the Bully-Victim group was significantly lower than the **Pure Bully** group (P < 0.01), it tells us that **not all perpetrators are the same.**

- **Pure Bullies (8.0%)** may have the confidence to manipulate others (Proactive Aggression).

- **Bully-Victims (15.0%)** lack confidence and likely use violence because they feel threatened or powerless (Reactive Aggression).

As visually summarized in Fig 2, there is clear decreasing trend in self-efficacy among adolescents involved in victimization and/or perpetration compared to non-involved peers.

## Factors independently associated with violence outcomes

**Multivariable determinants of violence involvement.** Multivariable binary logistic regression was utilized to isolate independent risk factors across the levels of the Socio-Ecological Model see Table 4.

The multivariable analysis identified distinct predictors for victimization and perpetration across all four SEM levels as shown in Table 4.

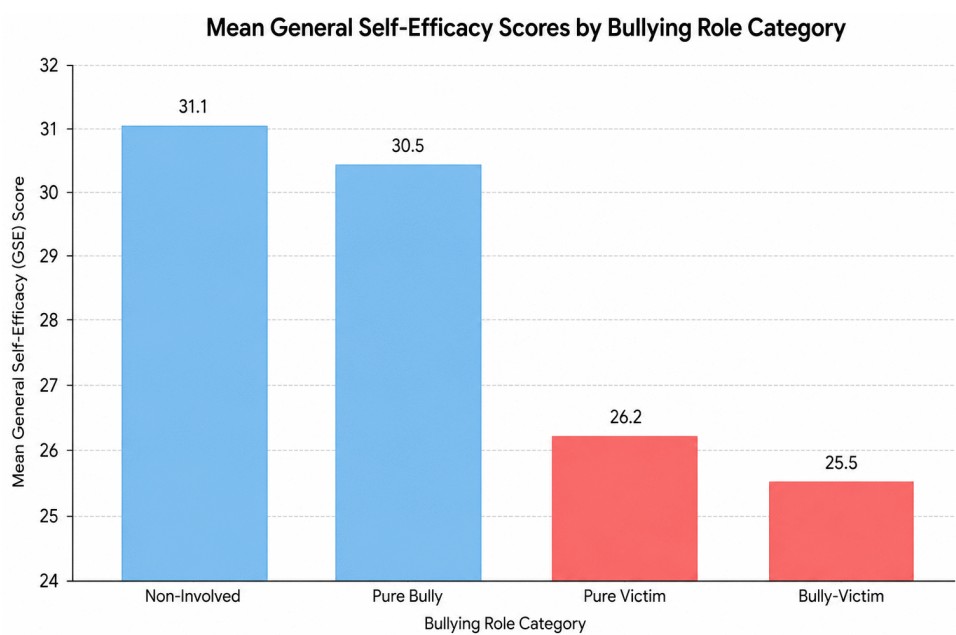

**Fig 2. Mean General self-efficacy scores by bullying role category among school adolescents in Asella, 2025 (N = 542).**

**Table 4. Multivariable logistic regression of factors associated with peer violence outcomes among adolescents in Asella, 2025.**

| Variable (SEM Level) | Victimization | | Perpetration | |
|---|---|---|---|---|
| | (COR [95% CI]) | (AOR [95% CI]) | (COR [95% CI]) | (AOR [95% CI]) |
| **Individual Level** | | | | |
| Low Self-Efficacy (≤28) | 3.31 [1.84, 5.95] | **2.88 [1.95, 4.26]*** | 2.03 [0.95, 4.33] | 1.12 [0.81, 1.55] |
| Male Gender | 1.74 [1.25, 2.42] | **1.91 [1.35, 2.69]*** | 2.50 [1.80, 3.48] | **2.35 [1.67, 3.30]*** |
| Age ≥ 15 Years | 1.30 [0.95, 1.78] | 1.15 [0.83,1.60] | 1.70 [1.25, 2.32] | **1.51 [1.10, 2.08]*** |
| **Relationship Level** | | | | |
| Non-Intact Parental Status | 1.55 [1.05, 2.28] | **1.47 [1.00, 2.16]*** | 1.38 [0.93, 2.04] | 1.25 [0.84, 1.86] |
| **Community Level** | | | | |
| Witnessing Community Violence | 1.43[0.66,3.09] | 1.08 [0.77, 1.52] | 2.78 [1.98, 3.90] | **2.52 [1.77, 3.59]*** |
| **Societal Level** | | | | |
| Lowest Wealth Quartile | 1.88 [1.32, 2.69] | **1.62 [1.11, 2.38]*** | 2.15 [1.50, 3.08] | **1.85 [1.28, 2.67]*** |

*Note: AOR = Adjusted Odds Ratio; CI = Confidence Interval. Asterisks () indicate statistical significance at P < 0.05.**

At the **Individual level**, low self-efficacy was the strongest predictor for being a victim of peer violence; adolescents with low GSE scores were nearly three times more likely to experience victimization compared to those with high scores (AOR = 2.88, 95% CI: 1.95–4.26). Male gender was a significant risk factor for both outcomes, doubling the odds for perpetration (AOR = 2.35). Interestingly, while older age (≥15 years) significantly increased the odds of perpetration (AOR = 1.51), it did not maintain significance for victimization after adjustment.

At the **Relationship level**, adolescents from non-intact households (divorced or separated parents) had 47% higher odds of victimization (AOR = 1.47).

At the **Community level**, witnessing violence in the neighborhood was the most powerful driver for engaging in perpetration; those exposed to community violence were 2.5 times more likely to act as perpetrators (AOR = 2.52, 95% CI: 1.77–3.59).

Finally, at the **Societal level**, economic status played a significant role; adolescents belonging to the lowest wealth quartile showed significantly increased odds for both victimization (AOR = 1.62) and perpetration (AOR = 1.85), highlighting the impact of structural poverty on school-based aggression.

## 4. Discussion

The findings of this study reveal a substantial burden of peer violence among school-attending adolescents in Asella, with **41.6%** experiencing victimization and **33.8%** engaging in perpetration within the last 12 months. These rates are consistent with the high prevalence of adolescent aggression reported across African region [5–7,31]. The results also align with evidence from Ethiopia, including studies from Debark [32] and Gondar [33], suggesting that peer violence remains a pervasive social and public health challenge in Ethiopian secondary schools.

### Individual level: The psychology of self-efficacy

At the individual level, low general self-efficacy emerged as the strongest predictor of victimization (**AOR = 2.88**; 95% CI: 1.95, 4.26). This finding strongly aligns with **Social Cognitive Theory (SCT)**, which posits that an individual's belief in their own agency determines how they navigate social stressors [12,13]. Adolescents with low self-efficacy may lack the assertive communication and problem-solving skills necessary to deter potential aggressors, thereby increasing their vulnerability to victimization.

A more nuanced understanding of this relationship is provided by the comparison of bullying roles. The **One-Way ANOVA** and subsequent **Tukey's HSD post-hoc tests** revealed that **Bully-Victims** (those involved in both roles) possessed the lowest mean self-efficacy (**25.5 ± 3.6**), which was significantly lower than that of **Pure Bullies** (**30.5 ± 4.5, P < 0.01**). This distinction suggests a divergence in aggressive motivations: while pure bullies may utilize **proactive aggression** backed by social confidence, bully-victims likely resort to **reactive aggression**. In this context, violence serves as a maladaptive defense mechanism for those who feel powerless to control their environment through non-violent means [33,34].

### Relationship and community levels: The cycle of violence

Our analysis identified **witnessing community violence** as the primary driver for perpetration (**AOR = 2.52**; 95% CI: 1.77, 3.59). This supports **Social Learning Theory**, suggesting that adolescents who observe violence in their neighbourhoods may internalize aggression as a normative and effective method for conflict resolution [33,42]. This "normalization" of violence bridges the gap between the community environment and individual behavioral choices.

Furthermore, **non-intact parental status** was found to be an independent predictor specifically for victimization (**AOR = 1.47**; 95% CI: 1.00, 2.16). The absence of a stable, two-parent household may limit the availability of emotional support and protective supervision, leaving adolescents more exposed to peer-group risks and less equipped with the resilience factors typically fostered in intact family units [9,22].

Interestingly, **age (≥15 years)** also emerged as a significant predictor for perpetration (**AOR = 1.51**; 95% CI: 1.10, 2.08). This suggests that as students in Asella progress into middle adolescence, they may face increased social pressure to adopt aggressive behaviors to navigate peer hierarchies or assert social dominance, a trend noted in similar urban Ethiopian settings [18,19].

### Societal level: Gender norms and economic status

The study highlights the role of broader societal structures in shaping violence. Male gender and lower household wealth were found to be robust predictors for both victimization and perpetration. This dual vulnerability underscores the role of structural socioeconomic disadvantage and rigid gender norms that often tolerate or encourage aggressive behavior among boys. Male

adolescents were significantly more likely to be both perpetrators (AOR = 2.35) and victims (AOR = 1.91), echoing global patterns [35,36]. This reflects rigid gender norms that often tolerate or encourage aggressive behavior among boys [37,38].

Finally, **socioeconomic disadvantage** played a critical role, with adolescents in the **lowest wealth quartile** facing higher odds of involvement in violence (Perpetration **AOR = 1.85**; Victimization **AOR = 1.62**). Structural poverty often limits access to extracurricular resources and creates high-stress environments, both of which are known to exacerbate peer conflict and reduce the perceived opportunity costs of engaging in risky behaviors [39,40].

### Synthesis of causal mechanisms within the SEM

When viewed through the Socio-Ecological Model, these findings reveal a "cascading risk" mechanism where distal societal factors and proximal community exposures converge to shape individual psychological states. The primary causal pathway for victimization appears to be the erosion of individual agency; structural poverty (Societal) and non-intact families (Relationship) may deplete the emotional resources necessary for an adolescent to develop high self-efficacy, leaving them vulnerable to peer-directed aggression. Conversely, the mechanism for perpetration is driven by the environmental normalization of aggression. Witnessing community violence (Community) acts as a powerful social primer, teaching adolescents that violence is a normative tool for navigating social hierarchies. This multi-level interaction creates a cycle where psychological vulnerability and environmental risk reinforce one another, suggesting that school-based violence in Asella cannot be resolved through individual counselling alone but requires disrupting the pathways between neighbourhood exposure and classroom behaviour.

### Strengths and limitations

A major strength of this study is the application of the **Socio-Ecological Model**, which allowed for the identification of multi-level factors beyond simple demographics. Additionally, the use of **Tukey's HSD post-hoc analysis** allowed for a rare psychological distinction between perpetrator subtypes, highlighting the unique vulnerability of "bully-victims" regarding self-efficacy. However, limitations must be acknowledged. First, the **cross-sectional design** precludes any definitive causal inferences: while we identified strong associations between low self-efficacy and victimization, we cannot determine if low self-efficacy is a precursor to or a consequence of violence exposure.

Second, despite the use of **self-administered questionnaires** to protect privacy, the reliance on self-reporting may still be subject to a degree of social desirability bias, potentially leading to an underreporting of perpetration. Third, as this study was conducted in the **urban setting of Asella**, the findings may not be fully generalizable to rural adolescent populations in Ethiopia, where social structures and peer dynamics may differ. Finally, this research focused on physical and relational aggression within the school environment and did not account for **cyber-bullying**, which represents an emerging dimension of peer violence in rapidly urbanizing areas.

## Conclusions

In conclusion, peer violence in Asella is a prevalent public health issue shaped by a complex interplay of individual psychological agency and environmental risk. While our study provides a robust multi-level snapshot of these dynamics, **the inherent limitations of the cross-sectional approach highlight the need for future longitudinal research** to track the long-term trajectory of these risk factors. Interventions should prioritize school-based programs that build self-efficacy alongside community-wide initiatives to de-normalize aggression. Addressing structural driver of poverty remains essential to disrupting the cycle of violence among school-going adolescents in Ethiopia.

### Recommendations

Based on the identified predictors across the socio-ecological levels, the following actions are recommended:

- **Individual Level (Psychological Resilience):** The Ministry of Education and local school administrations should integrate structured **Social-Cognitive Agency training into the school curriculum.** Programs should focus on enhancing **General Self-Efficacy,** assertive communication, and non-violent conflict resolution to empower potential victims and rehabilitate "Bully-Victims".

- **Relationship and Community Levels (Safety and Support):** Schools should establish peer-mentorship programs and strengthen school-family partnerships, particularly for adolescents from **non-intact households**. At the community level, peace-building initiatives are needed to reduce the normalization of violence that adolescents witness in their neighbourhoods.

- **Societal Level (Gender and Economy):** Implementation of **gender-transformative interventions** is necessary to challenge the harmful masculine norms that link manhood with aggression. Furthermore, government-led **poverty alleviation programs** targeting low-resource households may reduce the structural stressors that exacerbate school-level violence.

- **Future Research:** Longitudinal studies are recommended to establish the causal direction between psychological traits like self-efficacy and violence outcomes over time.

## Supporting information

**S1 File. Anonymized data set.**
(CSV)

## Acknowledgments

The authors would like to express their gratitude to Arsi University, College of Health Sciences, for providing the necessary facilities and administrative support to conduct this study. We also thank the Asella City Administration Education Bureau and the participating schools for their cooperation during the data collection process.

## Author contributions

**Conceptualization:** Ayalneh Demissie, Addis Wordofa, Muhammedawel Kaso.

**Data curation:** Ayalneh Demissie, Addis Wordofa, Muhammedawel Kaso.

**Formal analysis:** Ayalneh Demissie, Tesfa G/Meskel.

**Funding acquisition:** Ayalneh Demissie, Muhammedawel Kaso, Yehwalishet Demeke, Tahir Aman, Yonas Mulugeta.

**Investigation:** Ayalneh Demissie, Tesfa G/Meskel, Addis Wordofa, Muhammedawel Kaso, Yehwalishet Demeke.

**Methodology:** Ayalneh Demissie, Tesfa G/Meskel.

**Project administration:** Ayalneh Demissie, Addis Wordofa, Muhammedawel Kaso.

**Resources:** Ayalneh Demissie, Kasim Kimo, Addis Wordofa, Muhammedawel Kaso, Yehwalishet Demeke, Tahir Aman, Yonas Mulugeta.

**Software:** Ayalneh Demissie, Tesfa G/Meskel.

**Supervision:** Ayalneh Demissie, Kasim Kimo, Tesfa G/Meskel, Addis Wordofa, Tahir Aman.

**Validation:** Ayalneh Demissie, Kasim Kimo, Tesfa G/Meskel, Yonas Mulugeta.

**Visualization:** Ayalneh Demissie, Kasim Kimo, Tesfa G/Meskel, Yehwalishet Demeke, Tahir Aman, Yonas Mulugeta.

**Writing – original draft:** Ayalneh Demissie.

**Writing – review & editing:** Ayalneh Demissie.

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
