## [Decision Letter · Decision Letter 0]

3 Mar 2026

PONE-D-25-68551Subject: Submission of Original Research: "Individual, Community, and Societal Drivers of Adolescent Peer Violence in Asella, Ethiopia: A School-Based Cross-Sectional Study Utilizing the Socio-Ecological Model."

PLOS One

Dear Dr. Demissie,

Thank you for submitting your manuscript to PLOS ONE. After careful consideration, we feel that it has merit but does not fully meet PLOS ONE’s publication criteria as it currently stands. Therefore, we invite you to submit a revised version of the manuscript that addresses the points raised during the review process.

The manuscript has been evaluated by two reviewers, and their comments are available below and in the attached files.

Could you please carefully revise the manuscript to address all comments raised?

We look forward to receiving your revised manuscript.

Kind regards,

Ilse Bloom

Staff Editor

PLOS One

Journal Requirements:

[The authors declare no conflicts of interest.].

3. Thank you for stating the following in your manuscript:

[This research was financed by Arsi University, which supported data collection and field per diem for authors. The funder had no role in study design, data analysis, interpretation, writing, or the decision to submit the manuscript.]

[The authors declare no conflicts of interest.]

4. Please ensure that you refer to Figure 1 in your text as, if accepted, production will need this reference to link the reader to the figure.

5. We note you have included a table to which you do not refer in the text of your manuscript. Please ensure that you refer to Tables 2, 3, and 4 in your text; if accepted, production will need this reference to link the reader to the tables.

6. We note that there is identifying data in the Supporting Information file <Raw CVS data set.csv>. Due to the inclusion of these potentially identifying data, we have removed this file from your file inventory. Prior to sharing human research participant data, authors should consult with an ethics committee to ensure data are shared in accordance with participant consent and all applicable local laws.

-Location data

Please remove or anonymize all personal information (town names), ensure that the data shared are in accordance with participant consent, and re-upload a fully anonymized data set. Please note that spreadsheet columns with personal information must be removed and not hidden as all hidden columns will appear in the published file.

Reviewers' comments:

Reviewer's Responses to Questions

**Comments to the Author**

1. Is the manuscript technically sound, and do the data support the conclusions?

Reviewer #1: Yes

Reviewer #2: Yes

2. Has the statistical analysis been performed appropriately and rigorously?

Reviewer #1: Yes

Reviewer #2: No

3. Have the authors made all data underlying the findings in their manuscript fully available?

Reviewer #1: Yes

Reviewer #2: Yes

4. Is the manuscript presented in an intelligible fashion and written in standard English?

Reviewer #1: Yes

Reviewer #2: No

5. Review Comments to the Author

Reviewer #1: Dear Authors,

I have read your manuscript with great interest. I want to congratulate the authors on a well-conducted study. The study can constitute an important contribution to the empirical evidence on peer violence following the Socio-Ecological Model and the need for school and community-level initiatives to mitigate adolescent violence. However, I would like to draw the authors' attention to aspects of their manuscript that would need more detailed elaboration and consideration. Please refer to the attached document.

Reviewer #2: 1. In the abstract section, please include arguments that demonstrate the urgency and novelty of the study. The abstract is too dense with technical terms such as "Socio-Ecological Model (SEM)," "General Self-Efficacy (GSE)," and "bully-victims" without brief explanations, which can confuse the general reader.

2. The introduction presents issues that are too general and lack specificity regarding peer violence in urban settings like Asella. For example, there is no clear distinction between peer violence and other forms of violence, such as gender-based violence (GBV). Furthermore, there is no clear emphasis on urgency: no real-time data are available in Ethiopia, which would strengthen its relevance. The introduction may also be enhanced with clearer contextualization within secondary school education. The explanation of the underlying theory (socio-ecological model) could be more systematic, and the articulation of research gaps could be made more explicit.

3. In the methods section, the validity and reliability of the research instrument are not described in sufficient detail. In addition, the data collection procedures require further clarification. Inclusion criteria remain unclear; specify the exact age range and grade for participants to avoid selection bias, as older adolescents may have increased exposure to violence. Prevalence assumptions for sample size calculations are drawn from non-specific peer violence studies.

4. In the results section, please add narrative explanations for each table. The presentation of the results remains largely descriptive and lacks sufficient interpretive depth. The results do not display crude OR for comparison with adjusted OR (AOR), making it difficult to assess confounding factors.

5. The discussion section is too concise and normative, more like an extension of the conclusion than an in-depth analysis. The lack of exploration of causal mechanisms (e.g., how low self-efficacy mediates the effects of witnessing violence) makes it less compelling. Besides that, the linkage to the socio-ecological model needs to be strengthened. In addition, comparisons with previous studies are limited, and the implications for school health education are not discussed in sufficient depth.

6. In the conclusion section, please add research limitations.

6. PLOS authors have the option to publish the peer review history of their article (what does this mean?). If published, this will include your full peer review and any attached files.

Reviewer #1: No

Reviewer #2: No

---

## [Author Response · Author response to Decision Letter 1]

16 Apr 2026

Dear Dr. Bloom and Reviewers,

Thank you for the constructive feedback on our manuscript (PONE-D-25-68551). We have addressed all comments raised during the review process. Key revisions include:

Terminology Standardization: We have replaced "peer aggression" with "adolescent peer violence" and "peer victimization" throughout the manuscript to ensure academic and clinical precision.

Grammatical Improvements: We have addressed the subject-verb agreement and "verb confusion" errors noted in the text (e.g., clarifying the "structural drivers of poverty").

Data Privacy: We have provided a fully anonymized dataset (S1 Data) with all personal identifiers removed.

Funding Disclosure: The financial disclosure has been updated to clarify that Arsi University provided support for field per diems and data collection, but had no role in the study design or analysis.

A detailed, point-by-point response to every reviewer comment is attached as a separate PDF file labeled "Response to Reviewers."

Sincerely,

Dr. Ayalneh Demissie

---

## [Decision Letter · Decision Letter 1]

11 May 2026

Individual, Community, and Societal Drivers of Adolescent Peer Violence in Asella, Ethiopia: A School-Based Cross-Sectional Study Utilizing the Socio-Ecological Model

PONE-D-25-68551R1

Dear Dr. Demissie,

We’re pleased to inform you that your manuscript has been judged scientifically suitable for publication and will be formally accepted for publication once it meets all outstanding technical requirements.

Kind regards,

Vincenzo De Luca

Academic Editor

PLOS One

Additional Editor Comments (optional):

Reviewers' comments:

Reviewer's Responses to Questions

**Comments to the Author**

1. If the authors have adequately addressed your comments raised in a previous round of review and you feel that this manuscript is now acceptable for publication, you may indicate that here to bypass the “Comments to the Author” section, enter your conflict of interest statement in the “Confidential to Editor” section, and submit your "Accept" recommendation.

Reviewer #1: All comments have been addressed

Reviewer #2: All comments have been addressed

2. Is the manuscript technically sound, and do the data support the conclusions?

Reviewer #1: Yes

Reviewer #2: Yes

3. Has the statistical analysis been performed appropriately and rigorously?

Reviewer #1: Yes

Reviewer #2: Yes

4. Have the authors made all data underlying the findings in their manuscript fully available?

Reviewer #1: Yes

Reviewer #2: Yes

5. Is the manuscript presented in an intelligible fashion and written in standard English?

Reviewer #1: Yes

Reviewer #2: Yes

6. Review Comments to the Author

Reviewer #1: (No Response)

Reviewer #2: This manuscript addresses an interesting and relevant topic, and the research question is generally clear. The authors have addressed all of the reviewers' feedback. These improvements make this study potentially useful to readers in this field, particularly as it seeks to provide evidence on an issue of practical and scientific importance.

7. PLOS authors have the option to publish the peer review history of their article (what does this mean?). If published, this will include your full peer review and any attached files.

Reviewer #1: No

Reviewer #2: No

---

## [Editor Report · Acceptance letter]

PONE-D-25-68551R1

PLOS One

Dear Dr. Demissie,

I'm pleased to inform you that your manuscript has been deemed suitable for publication in PLOS One. Congratulations! Your manuscript is now being handed over to our production team.

Kind regards,

on behalf of

Dr. Vincenzo De Luca

Academic Editor

PLOS One